# Sulcal morphology of posteromedial cortex substantially differs between humans and chimpanzees

Ethan H. Willbrand [1,2,4], Samira A. Maboudian [2,4], Joseph P. Kelly[1], Benjamin J. Parker[2], Brett L. Foster[3] & Kevin S. Weiner [1,2✉]

Recent studies identify a surprising coupling between evolutionarily new sulci and the functional organization of human posteromedial cortex (PMC). Yet, no study has compared this modern PMC sulcal patterning between humans and non-human hominoids. To fill this gap in knowledge, we first manually defined over 2500 PMC sulci in 120 chimpanzee (*Pan Troglodytes*) hemispheres and 144 human hemispheres. We uncovered four new sulci, and quantitatively identified species differences in sulcal incidence, depth, and surface area. Interestingly, some sulci are more common in humans and others, in chimpanzees. Further, we found that the prominent marginal ramus of the cingulate sulcus differs significantly between species. Contrary to classic observations, the present results reveal that the surface anatomy of PMC substantially differs between humans and chimpanzees—findings which lay a foundation for better understanding the evolution of neuroanatomical-functional and neuroanatomical-behavioral relationships in this highly expanded region of the human cerebral cortex.

[1] Department of Psychology, University of California Berkeley, Berkeley, CA 94720, USA. [2] Helen Wills Neuroscience Institute, University of California Berkeley, Berkeley, CA 94720, USA. [3] Department of Neurosurgery, Perelman School of Medicine, University of Pennsylvania, Philadelphia, PA 19104, USA. [4]These authors contributed equally: Ethan H. Willbrand, Samira A. Maboudian. ✉email: kweiner@berkeley.edu

A fundamental question in comparative biology and systems neuroscience is: What features of the brain are unique to humans? Key insights regarding what features of the brain are human-specific have been gleaned from studies comparing anatomical and functional features of the human brain to features from the brains of our close evolutionary relative, the chimpanzee[1–25]. Of all the features to study, researchers particularly focus on the folds of the cerebral cortex, or sulci, as they generally track with evolutionary complexity[26]. For example, while mice and marmosets have rather smooth, lissencephalic cerebral cortices, 60–70% of the folded, gyrencephalic cerebral cortex of hominoids is buried within sulci[4,27]. Intriguingly, recent studies have identified "evolutionarily new" shallow sulci that have been linked to functional organization across a broad array of cognitive domains (e.g.,[18,23,28–43]), several of which reflect cognitive abilities that are arguably unique to humans. We refer to these small, shallow, and variable sulci as "evolutionarily new" because they are located either in association cortices that have expanded throughout evolution (e.g., lateral prefrontal cortex, medial parietal cortex, etc.[3,10,12–14,16,44–46]) or are unique to the hominoid brain and are absent in other primates (e.g., the fusiform gyrus[19,47]; Materials and Methods for additional details on sulcal classification). Building on this previous work, we compared the sulcal patterning of the posteromedial cortex (PMC)—a region on the medial cortical surface that includes the posterior cingulate, retrosplenial, and precuneal cortices[24,48–50]—between humans and chimpanzees with a particular emphasis on the smaller, shallower, and relatively overlooked "evolutionarily new" cortical indentations[24,25].

The sulcal organization of PMC has been under-documented, even in the most recent neuroanatomical treatises (e.g.,[51,52]). Nevertheless, PMC is critically important in hominoids as it contains regions implicated in the default mode and cognitive control networks[49,53–57] with complex structural and functional connections[48,49,55,58]. PMC is also implicated in many complex cognitive abilities[49,54,59–62] and is particularly susceptible to neurodegenerative disease[60]. Thus, quantifying the similarities and differences in the PMC sulcal patterning between chimpanzees and humans will not only shed light on the comparative neuroanatomy of PMC between species, but also provide understanding regarding structural-functional relationships between species with potential cognitive insights[25].

While it is known that the larger (primary) sulci within PMC are present in chimpanzees[63–65] and the inframarginal sulcus—a newly uncovered smaller PMC sulcus—is variably present in chimpanzees[24], the phylogenetic emergence of a majority of recently clarified PMC sulci[24] has yet to be compared between chimpanzees and humans. Therefore, in the present study, we comprehensively examined the PMC sulcal patterning between humans and chimpanzees using cortical surface reconstructions as in our prior work[19,24,43]. Our analyses were guided by three main questions. First, does the amount of PMC buried in sulci differ between humans and chimpanzees? Second, do the incidence rates of PMC sulci differ between species? Third, do the primary morphological features of these structures (i.e., depth and surface area) differ between species? Here, we uncovered four new sulci, and quantitatively identified species differences in incidence rates, depth, and surface area. Interestingly, some PMC sulci are more common in humans and others, in chimpanzees. Further, we found that the prominent and consistent marginal ramus of the cingulate sulcus differs significantly between species.

## Results

In order to answer these main questions, we examined the PMC of 72 young adult humans [from the Human Connectome Project (HCP; http://www.humanconnectomeproject.org/)] and 60 chimpanzees [from the National Chimpanzee Brain Resource (https://www.chimpanzeebrain.org/)]. These participants were used in prior work to assess the anatomical, functional, and evolutionary significance of a new tripartite landmark in PCC, the inframarginal sulcus (ifrms[24]), but the rest of the PMC sulci were not considered in these previous cross-species analyses until the present study.

To broadly determine how much of the PMC is sulcal vs. gyral in each species, we calculated how much of the regions corresponding to an automated parcellation of PMC in FreeSurfer (v6.0.0; surfer.nmr.mgh.harvard.edu)[66] were buried in sulci (i.e., the percentage of vertices with values above zero in the .sulc file[67]) via the Dice coefficient (Fig. 1a; Materials and

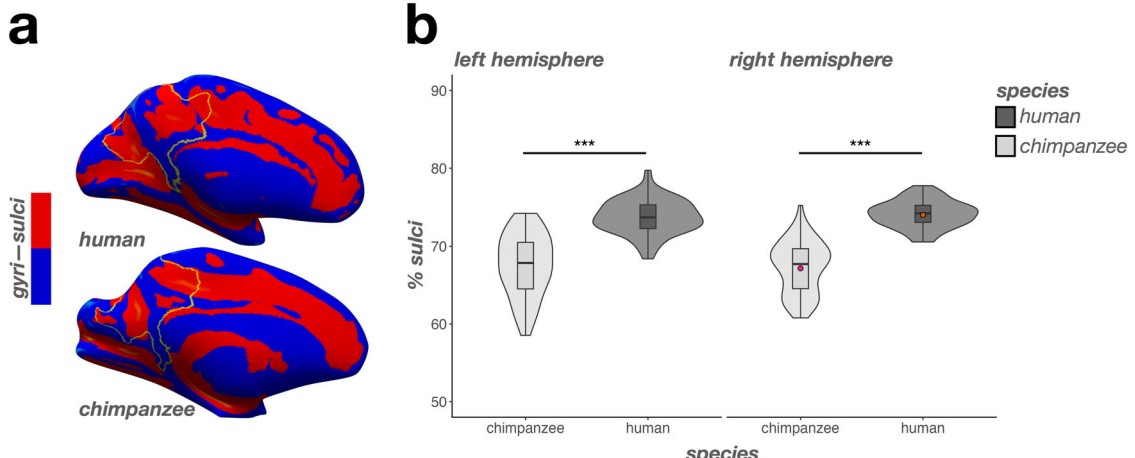

**Fig. 1 The percentage of PMC buried in sulci differs between humans and chimpanzees. a** Inflated human (top) and chimpanzee (bottom) right hemisphere cortical surface reconstructions (mirrored for visualization purposes). The outline of automatically defined PMC from the Destrieux parcellation[66] is indicated in yellow. The FreeSurfer.sulc file[67] is overlaid on each surface (Sulci: red; Gyri: blue). These individual surfaces present the average PMC sulcation for each species (Human: 73.9%; Chimpanzee: 67.4%), which are represented as colored dots in **b**. **b** Violin plots (box plot and kernel density estimate) visualizing the percentage of PMC in sulci (percentage values are out of 100) as a function of species (x-axis) and hemisphere (left: left hemisphere; right: right hemisphere). The significant difference in PMC sulcation between species (as a result of the main effect of species) is indicated with asterisks (***p < 0.001).

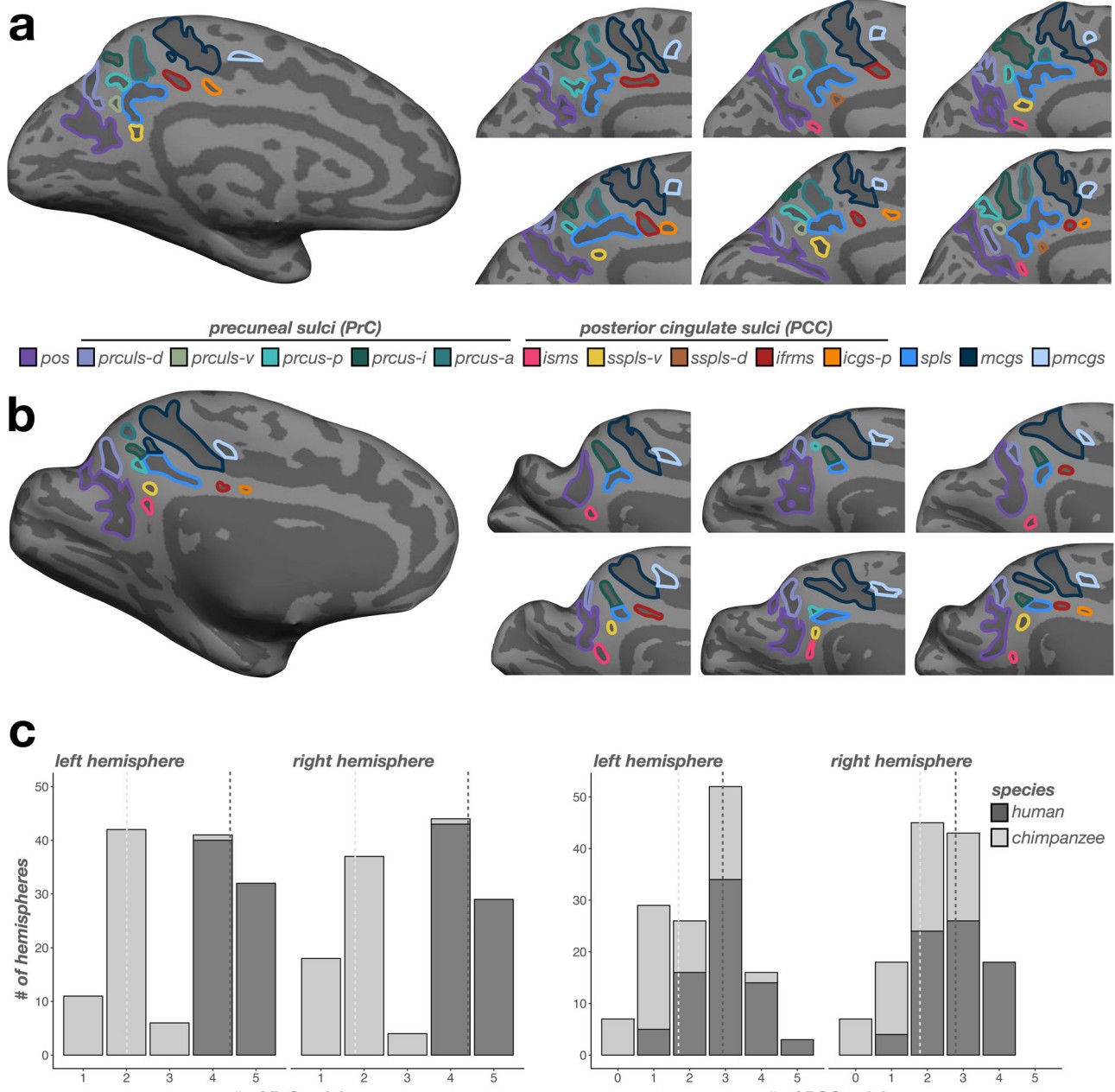

**Fig. 2 Humans have more PMC sulci than chimpanzees across hemispheres in both PrC and PCC. a** Left: An inflated cortical surface reconstruction of an individual human hemisphere. Sulci: dark gray; Gyri: light gray. Individual posteromedial (PMC) sulci are outlined according to the legend at the bottom. Right: Six example hemispheres zoomed in on the PMC depicting variations of sulcal incidence between participants. Right hemisphere images are mirrored so that all images have the same orientation. **b** Same as **a**, but for chimpanzee hemispheres. **c** Left: Incidence rates of precuneal (PrC) sulci (x-axis; see legend in a) across species (colors, see legend) for each hemisphere (left: left hemisphere; right: right hemisphere). Dashed lines indicate the average number of sulci for each species in each hemisphere. Right: Same as the left, but for posterior cingulate (PCC) sulci.

Methods). Replicating prior postmortem work[4,27,68], the majority of human PMC was buried in sulci (mean±std = 73.9 ± 1.97%). Chimpanzee PMC was relatively less sulcated (mean±std = 67.4 ± 3.69%; Fig. 1b). A linear mixed effects model (LME) with factors of species and hemisphere (controlling for differences in brain size), confirmed this large difference between species (main effect of species: F(1, 130) = 220.57, p < .0001, η2 = 0.63; Fig. 1b). There were no hemispheric differences (ps > 0.24).

Next, we manually defined sulci in precuneal (PrC) and posterior cingulate cortices (PCC) — which are subregions of the PMC[24,48–50]—in all human and chimpanzee brains (Materials and Methods for a detailed description of these sulci). All PMC

sulci were defined on cortical reconstructions from FreeSurfer (Fig. 2a, b for example human and chimpanzee hemispheres; Supplementary Figs. 1, 2 for all human and chimpanzee hemispheres). Once all sulci were defined, we quantified the incidence, average sulcal depth (normalized to the max depth in each hemisphere) and surface area (normalized to the total surface area of each hemisphere) of each PMC sulcus (Materials and Methods).

We began by quantifying the incidence rates of PMC-related sulci in three groups: i) sulci that serve as the bounding perimeter of PMC or delimit PMC subregions (PrC, PCC), ii) sulci within PrC, and iii) sulci within PCC. Crucially, this procedure revealed

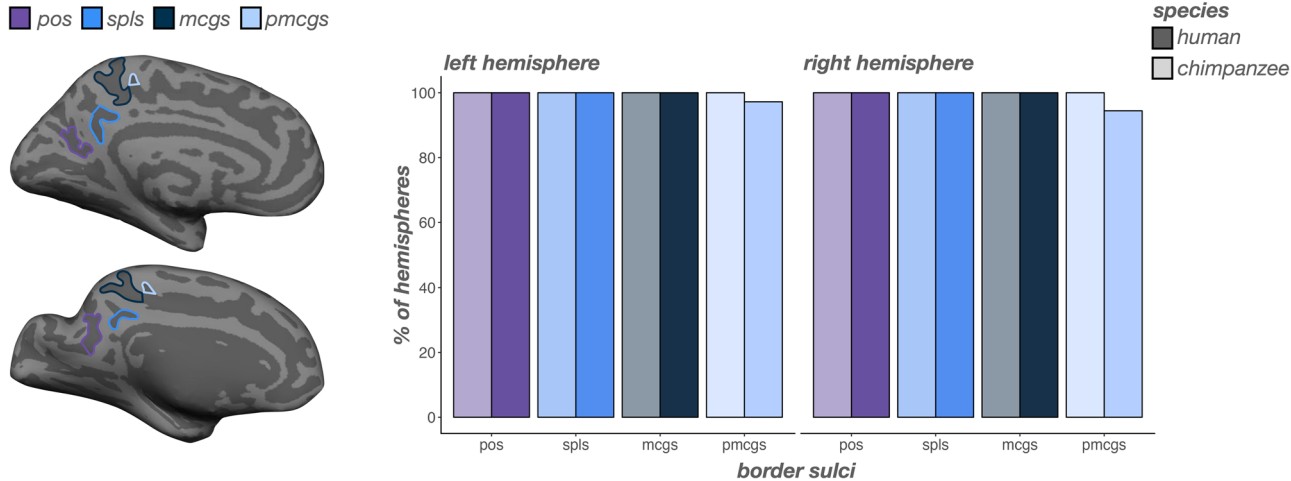

**Fig. 3 Incidence rates of sulci that delimit PMC and its subregions are comparable between humans and chimpanzees.** Left: An inflated cortical surface reconstruction of an individual human (top) and chimpanzee (bottom) hemisphere with sulci that bound PMC and its subregions (PrC, PCC) outlined according to the legend at the top of the figure. Right: Bar plots visualizing incidence rates (percent of hemispheres) as a function of sulcus (x-axis), species (darker colors: human; lighter colors: chimpanzee), and hemisphere (left: left hemisphere; right: right hemisphere). Sulci are generally ordered posterior to anterior.

four new sulci that were not considered in prior work of PMC sulcal morphology (e.g.[24,55,64,68–72], Fig. 2a, b; Supplementary Figs. 3, 4). While we labeled and quantified the incidence rates of these four sulci across species for the first time to the best of our knowledge, some present and modern anatomists often included an unlabeled sulcus in the location of some these sulci in their summary schematics (Supplementary Figs. 3, 4). Further, these sulci were identifiable in postmortem chimpanzee hemispheres from a classic neuroanatomical atlas[65], indicating that Free-Surfer's computational processes did not artificially create shallow sulci (Supplementary Fig. 4). We described across-species comparisons for each group in turn below using logistic regression GLMs with species (human, chimpanzee) and hemisphere (left, right), as well as their interaction, as factors for sulcal presence. Afterwards, we compared the depth and surface area of PMC sulci between species using LMEs with species (human, chimpanzee), sulcus (PMC sulci), and hemisphere (left, right), as well as their interaction, as factors. Finally, we repeat these analyses on the incidence and morphology of the marginal ramus of the cingulate sulcus—a prominent sulcal landmark in PMC[24,55,64,68–72] that, contrary to previous studies, differs substantially between species, which we show here. In all analyses, observed hemispheric asymmetries were not significant (ps > 0.05), unless explicitly stated. Further, for all post hoc comparisons conducted, p-values were corrected with Tukey's methods (Materials and Methods for additional analysis details).

**Incidence rates of large and deep sulci that delimit PMC and its subregions do not differ across species.** We identified two large and deep sulci serving as anterior and posterior bounds of PMC respectively, the marginal ramus of the cingulate sulcus (mcgs) and parieto-occipital sulcus (pos), as well as the splenial sulcus (spls), which separates PrC from PCC (Fig. 2a, b; Materials and Methods). Replicating prior post-mortem work[63–65], we found that the mcgs, spls, and pos were present in all humans and chimpanzees (Fig. 3). We also identified a consistent sulcus just anterior to the mcgs (Fig. 2a, b). As it is common practice to refer to consistent sulci as "pre" or "post" if they are located in front of or behind other prominent sulci (e.g., precentral, postcentral, and central sulci), we refer to this sulcus as the premarginal branch of the cingulate sulcus (pmcgs). When present, the pmcgs is located

just under the paracentral fossa and serves as the point where the mcgs breaks from the cingulate sulcus (cgs) proper. Even though the pmcgs is technically located outside of what is considered PMC, we include it here as (to our knowledge), its consistency across hemispheres and species has not yet been documented until the present work (Materials and Methods for additional information regarding the pmcgs; Supplementary Fig. 5). The pmcgs was clearly identifiable in 97.22% of left and 94.4% of right hemispheres in humans and in 100% of chimpanzees (Fig. 3). The incidence rates for these four sulci were comparable between species (no main effect of species: $\chi2(1) = 2.45$, $p = 0.12$; Fig. 3).

**Incidence rates of PrC sulci differ substantially across species, including the newly identified ventral precuneal limiting sulcus (prculs-v).** In human PrC, the posterior (prcus-p), intermediate (prcus-i), and anterior precuneal sulci (prcus-a), as well as the dorsal precuneal limiting sulcus (prculs-d) were present in all hemispheres (Figs. 2a, 4). Previously, we referred to this latter sulcus as the prculs[24] (mirroring the label from a recent neuroanatomical atlas[69]). However, here, we also consistently identified a ventral sulcal component in a comparable posterior plane as the prculs-d, but more inferiorly situated between the prculs-d and the spls (Fig. 2a). Consequently, we refer to this sulcus as the ventral prculs (prculs-v), which was identifiable in 44.44% of left and 40.28% of right hemispheres in humans (Fig. 4).

In contrast, PrC sulci were far more variable in chimpanzees (Fig. 2b, c). Generally, humans contained more sulci than chimpanzees in PrC (F(1, 130) = 1194.13, $p < .0001$, $\eta2 = 0.90$; Fig. 2c, left). The prculs-d was the only sulcus comparably present between species (left: 96.67%; right: 96.67%; no main effect of species: $\chi2(1) = 3.19$, $p = 0.07$; Fig. 4). Interestingly, among the three recently identified prcus components[24], prcus-i was the second most present PrC sulcus in chimpanzees, but was still less present than in humans (left: 76.67%; right: 73.33%; main effect of species: $\chi2(1) = 24.09$, $p < 0.0001$; Fig. 4). Conversely, prcus-p (left: 15%; right: 5%; main effect of species: $\chi2(1) = 125.39$, $p < 0.0001$) and prcus-a (left: 6.67%; right: 5%; main effect of species: $\chi2(1) = 150.56$, $p < .0001$) were quite rare in chimpanzees (Fig. 4). Finally, the newly identified prculs-v in humans was not identifiable in any chimpanzee hemispheres examined (main effect of species: $\chi2(1) = 47.30$, $p < .0001$; Fig. 4).

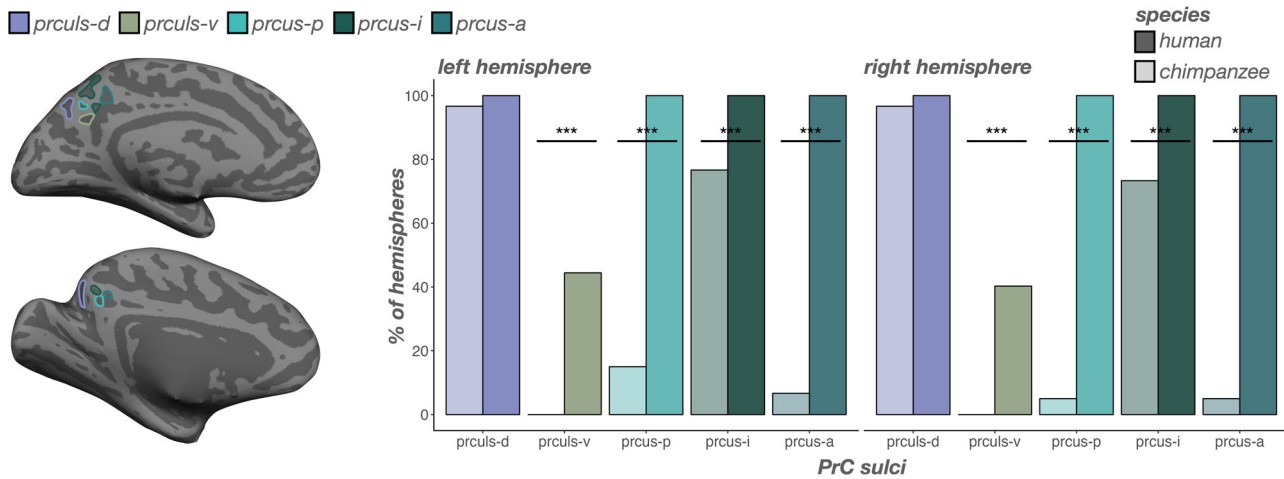

**Fig. 4 Incidence rates of precuneal (PrC) sulci are generally higher in humans than chimpanzees.** Left: An inflated cortical surface reconstruction of an individual human (top) and chimpanzee (bottom) hemisphere with PrC sulci outlined according to the legend. Right: Bar plots visualizing incidence rates (percent of hemispheres) as a function of sulcus (x-axis), species (darker colors: human; lighter colors: chimpanzee), and hemisphere (left: left hemisphere; right: right hemisphere). Sulci are generally ordered posterior to anterior. Lines and asterisks highlight significant differences in incidence between species (*p < .05, ***p < .001). The intermediate precuneal sulcus (prcus-i) is the most common of the three precuneal sulci in chimpanzees. In comparison to the consistency of the prcus-i, prcus-a and prcus-p are extremely rare in chimpanzees.

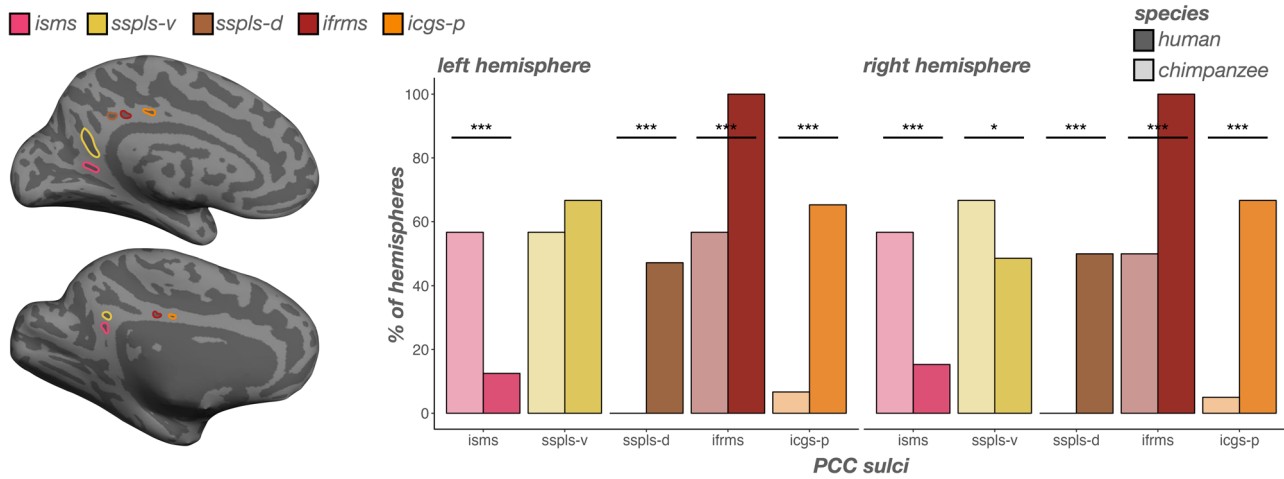

**Fig. 5 Incidence rates of posterior cingulate (PCC) sulci are variable between humans and chimpanzees.** Left: An inflated cortical surface reconstruction of an individual human (top) and chimpanzee (bottom) hemisphere with PCC sulci outlined according to the legend. Right: Bar plots visualizing incidence rates (percent of hemispheres) as a function of sulcus (x-axis), species (darker colors: human; lighter colors: chimpanzee), and hemisphere (left: left hemisphere; right: right hemisphere). Sulci are generally ordered posterior to anterior. Lines and asterisks highlight significant differences in incidence between species (*p < .05, ***p < .001). The isms and sspls-v are more common in chimpanzees than humans. The sspls-d, ifrms, and icgs-p are more common in humans than chimpanzees. ifrms data from ref. [24].

**Incidence rates of PCC sulci differ substantially across species, in which the newly identified ventral subsplenial sulcus (sspls-v) and isthmus sulcus (isms) are identifiable as frequently or more frequently in chimpanzees than humans.** Sulci in human PCC are more variable than those in human PrC (Fig. 2a, c)[24]. Generally, humans contained more sulci in PCC than chimpanzees (F(1, 130) = 63.86, p < 0.0001, η2 = 0.33; Fig. 2c, right). As shown previously, the inframarginal sulcus (ifrms) is the only PCC sulcus present in 100% of human hemispheres (Fig. 5)[24]. The ifrms is identifiable in 50% of chimpanzee hemispheres (Fig. 5)[24]. Anterior to the ifrms, the posterior intracingulate sulcus (icgs-p) was present in 65.28% of left and 66.67% right hemispheres in humans, and rarely identifiable in chimpanzees (*left*: 6.67%; *right*: 5%; main effect of species: χ2(1) = 53.74, p < 0.0001; Fig. 5). Posterior to the ifrms, the dorsal subsplenial sulcus (sspls-d) was present in 47.22% of left and 50% right

hemispheres in humans, and was not identifiable in any chimpanzee hemispheres (main effect of species: χ2(1) = 51.02, p < 0.0001; Fig. 5).

While we previously referred to the sspls-d as the sspls[24], here, we also identified an additional sulcus that was consistently identifiable just ventral and discontinuous with the dorsal component (Fig. 2a, b). As such, we refer to this newly-identified sulcus as the ventral sspls (sspls-v), which in humans was present in 66.67% of left hemispheres and 48.61% of right hemispheres (Fig. 5). Interestingly, the sspls-v showed no main effect of species (χ2(1) = 1.39, p = 0.24), but an interaction between species and hemisphere (χ2(1) = 5.34, p = 0.02), such that in chimpanzees, it was present in a comparable amount of left hemispheres to humans (56.67%; p = 0.24), but was present in more chimpanzee right hemispheres than human right hemispheres (66.67%; odds ratio = 0.75, p = 0.03; Fig. 5).

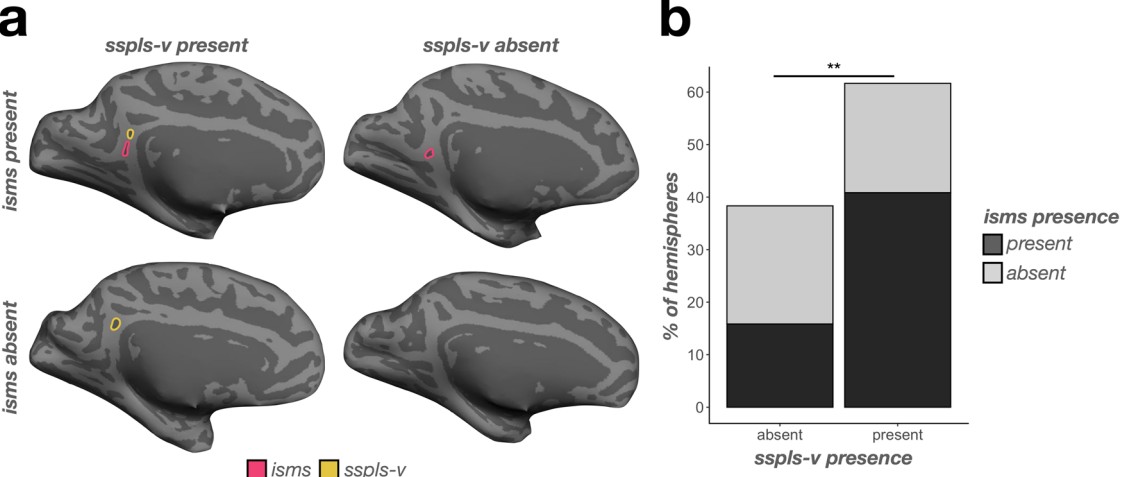

**Fig. 6 Incidence of the sspls-v is related to the incidence of the isms in chimpanzees. a** Four example inflated chimpanzee hemispheres displaying the four combinations of sspls-v (outlined in yellow when present) and isms (outlined in pink when present): both present (top left), sspls-v present (bottom left), isms present (top right), and both absent (bottom right). **b** Bar plot visualizing the frequency of sspls-v and isms presence (colors, see legend). When the sspls-v is present, the isms is more likely present rather than absent; when the sspls-v is absent, the isms is likely to be absent (**$p < .01$).

Finally, in a minority of humans (12.50% of left and 15.28% of right hemispheres), we could identify a previously undefined sulcus inferior to the sspls-v within the isthmus of the cingulate gyrus, which we termed the isthmus sulcus (isms; Figs. 2a, 5). The isms was present in more chimpanzee hemispheres (56.67% of left and right hemispheres) than humans (main effect of species: $\chi2(1) = 30.26$, $p < .0001$; Fig. 5). Interestingly, the incidence of the two more common PCC sulci in chimpanzees (sspls-v and isms) were related in chimpanzees ($\chi2(1) = 7.01$, $p = 0.008$), such that chimpanzees with an sspls-v were more likely to have an isms (odds ratio = 4.77; Fig. 6). No other sulcal incidence rates were related ($ps > 0.10$). To further summarize these relationships, there was a PCC region (dorsal PCC, ventral PCC) and species interaction on sulcal presence ($\chi2(1) = 74.79$, $p < 0.0001$), where post hoc comparisons showed that, overall, dorsal PCC sulci (sspls-d, ifrms, icgs-p) were less common in chimpanzees than humans (odds ratio = −2.33, $p < 0.0001$), whereas ventral PCC sulci (isms, sspls-v) were more common in chimpanzees than humans (odds ratio = 0.96, $p < 0.0001$; Fig. 5).

**The relative depth and surface area of PMC sulci largely differ between chimpanzees and humans.** In terms of depth, an LME with predictors of sulcus, hemisphere, and species revealed three species-related findings. First, a main effect of species (F(1, 130) = 269.48, $p < 0.0001$, η2 = 0.67) showed that human PMC sulci were relatively deeper than chimpanzees (Fig. 7a). Second, an interaction between species and sulcus (F(7, 1497) = 131.81, $p < 0.0001$, η2 = 0.38) indicated more complex relationships at the individual-sulcus level. Post hoc analyses revealed three findings: i) the isms, pos, prculs-d, prcus-i, spls, and sspls-v were relatively deeper in humans than chimpanzees ($ps < 0.003$), ii) the mcgs was relatively deeper in chimpanzees than humans ($p = 0.04$), and iii) the pmcgs was comparably deep between species ($p = 0.45$; Fig. 7a). Third, a three-way interaction among species, sulcus, and hemisphere (F(7, 1497) = 2.43, $p = 0.01$, η2 = 0.01) showed that the mcgs was deeper in chimpanzees in the left hemisphere ($p = 0.04$), but comparably deep in the right hemisphere ($p = 0.35$; Fig. 7a) compared to humans.

In terms of surface area, an LME with predictors of sulcus, hemisphere, and species also revealed three species-related findings. First, a main effect of species (F(1, 130) = 6.51, $p = 0.01$, η2 = 0.05) showed that human PMC sulci were

relatively larger than chimpanzees (Fig. 7b). Second, an interaction between species and sulcus (F(7, 1497) = 70.67, $p < 0.0001$, η2 = 0.25) indicated that the latter main effect was driven by differences at an individual-sulcus level. Post hoc analyses revealed three findings: i) the spls, prculs-d, and prcus-i were relatively larger in humans than chimpanzees ($ps < 0.0001$), ii) the pos, mcgs, and pmcgs were relatively larger in chimpanzees than humans ($ps < 0.02$), and iii) the isms and sspls-v were comparably large between species ($ps > 0.62$; Fig. 7b). Third, a three-way interaction among species, sulcus, and hemisphere (F(7, 1497) = 8.65, $p < 0.0001$, η2 = 0.04) showed that: i) the species difference for prculs-d was larger in the left hemisphere (estimate = −0.0015, $p < 0.0001$) than the right (estimate = −0.0008, $p = 0.01$), ii) the pmcgs was marginally relatively larger in chimpanzees in the left hemisphere ($p = 0.05$) but not the right hemisphere ($p = 0.18$), and iii) the pos is relatively larger in chimpanzees in the left hemisphere ($p < 0.0001$), but not the right hemisphere ($p = 0.24$; Fig. 7b).

**Morphological types of the mcgs differ substantially between humans and chimpanzees.** Previous work by Bailey and colleagues[64] showed that the chimpanzee mcgs bifurcated into what they termed "vertical" and "horizontal" components. Conversely, Ono and colleagues[70] identified that the human mcgs could variably present with side branches and/or a bifurcated dorsal end. In the present study, we integrated these previous classifications into four patterns based on what branches were present. We could identify up to three different branches of the mcgs: i) the main branch (mb) extending from the cingulate sulcus, ii) a branch extending dorsally from the main branch (db), and iii) a side branch (sb) extending horizontally or ventrally from the main branch (termed cih by Bailey et al.[64]). In the neuroanatomical literature, it is common to qualitatively describe sulcal "types" based on variation in the shape of a given sulcus and/or patterning of fractionation or intersection with neighboring sulci (e.g.,[37,73–75]). Following this terminology, the combination of these branches fell into four types: I) mb with no db or sb, II) mb with a db, III) mb with a sb, and IV) mb with both a db and sb (Fig. 8a, b).

We quantitatively determined whether the incidence rates of the four mcgs types differed by species, as well as between hemispheres for each species with $\chi2$ tests. We observed

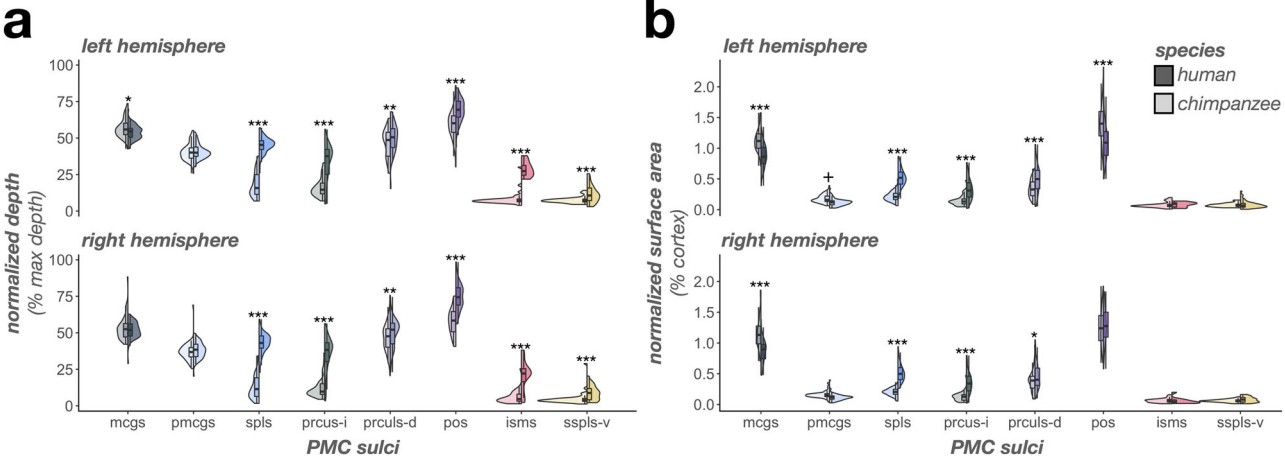

**Fig. 7 The complex relationship of PMC sulcal morphology in humans versus chimpanzees. a** Split violin plots (box plot and kernel density estimate) visualizing normalized sulcal depth (percent of max depth; percentage values are out of 100) as a function of sulcus (x-axis), species (darker colors, right violin: human; lighter colors, left violin: chimpanzee), and hemisphere (top: left hemisphere; bottom: right hemisphere). Significant differences between species [as a result of the species x sulcus interaction (or the species x sulcus x hemisphere interaction for the mcgs)] are indicated with asterisks ($^{*}p < 0.05$, $^{**}p < 0.01$, $^{***}p < 0.001$). **b** Same as a, but for normalized surface area (percent of cortical surface area; percentage values are out of 100). Significant differences between species [as a result of the species x sulcus interaction (or the species x sulcus x hemisphere interaction for the prculs-d, pmcgs, and pos)] are indicated with asterisks ($^{+}p = 0.05$; $^{*}p < 0.05$, $^{**}p < 0.01$, $^{***}p < 0.001$).

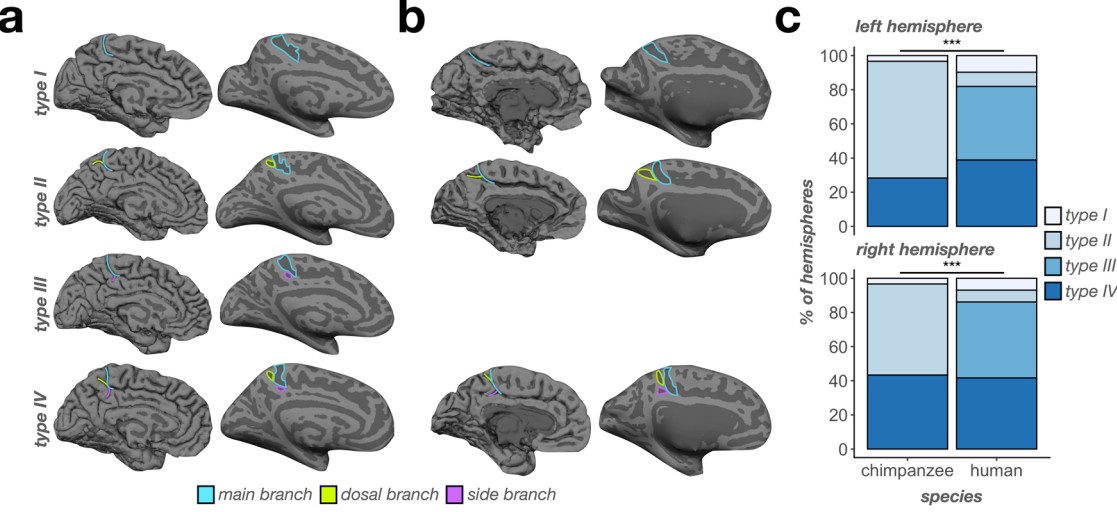

**Fig. 8 Chimpanzees do not have a Type III mcgs. a** Example pial (left) and inflated (right) human hemispheres displaying the four "types" of the mcgs. Type I consists of only a main branch (blue outlines/lines). Type II consists of a main branch and a dorsal branch (green outlines/lines). Type III consists of a main branch and a side branch (purple outlines/lines). Type IV consists of all three branches. **b** Same as **a**, but for chimpanzees. Note that no chimpanzees in our sample had an identifiable type III mcgs (empty third row). **c** Bar plot visualizing the incidence of mcgs types as a function of species (x-axis), type (color, see legend), and hemisphere (top: left hemisphere; bottom: right hemisphere). Lines and asterisks highlight significant species differences in the incidence of mcgs types in both hemispheres ($^{***}p < .001$).

significant differences in both hemispheres (*left*: $\chi2(3) = 61.95$, $p < 0.0001$; *right*: $\chi2(3) = 52.62$, $p < 0.0001$; Fig. 8c). Specifically, type I was comparably present between species in both the left ($p = 0.24$; *chimpanzee*: 3.33%; *human*: 9.72%) and right hemispheres ($p = 0.34$; *chimpanzee*: 3.33%; *human*: 6.94%; Fig. 8c). Type II was more present in chimpanzees (and the most common type) than humans in both the left ($p < 0.0001$; *chimpanzee*: 68.33%; *human*: 8.33%) and right hemispheres ($p < 0.0001$; *chimpanzee*: 53.33%; *human*: 6.94%; Fig. 8c). Conversely, type III was only present in humans (and the most common type) in both the left ($p < 0.0001$; *chimpanzee*: 0%; *human*: 43.06%) and right hemispheres ($p < 0.0001$; *chimpanzee*: 0%; *human*: 44.44%; Fig. 8c). Finally, type IV was equally present in both the left ($p = 0.13$; *chimpanzee*: 28.33%; *human*: 38.89%) and right

hemispheres ($p = 0.69$; *chimpanzee*: 43.33%; *human*: 41.67%; Fig. 8c) across species.

**The depth and surface area of mcgs components largely differ between chimpanzees and humans.** Finally, we quantitatively tested for species differences in the sulcal depth and surface area of the three mcgs components comprising the different types (mb, db, and sb). In terms of depth, an LME with predictors of component, hemisphere, and species on mcgs component sulcal depth revealed five findings. First, there was a main effect of component (F(2, 341) = 440.90, $p < 0.0001$, $\eta2 = 0.72$), such that the mb was deeper than the db and sb ($ps < 0.0001$) and the db was deeper than the sb ($p < 0.0001$; Fig. 9a). Second, there was a main effect of hemisphere (F(1, 130) = 25.25, $p < 0.0001$,

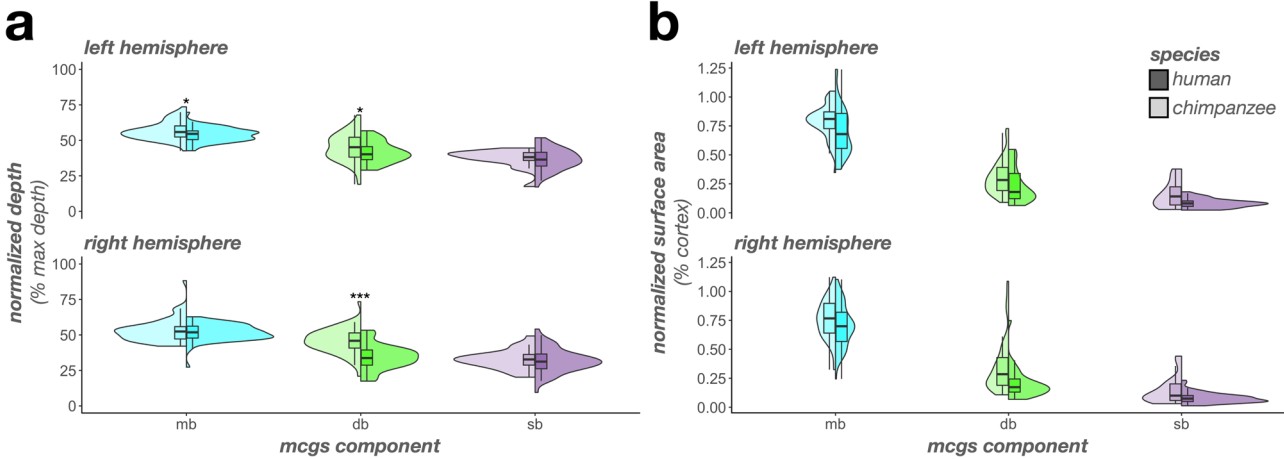

**Fig. 9 The mcgs is morphologically distinct between humans and chimpanzees. a** Split violin plots (box plot and kernel density estimate) visualizing normalized sulcal depth (percent of max depth; percentage values are out of 100) as a function of mcgs component (x-axis), species (darker colors, right violin: human; lighter colors, left violin: chimpanzee), and hemisphere (top: left hemisphere; bottom: right hemisphere). Significant differences between species (as a result of the species x component x hemisphere interaction) are indicated with asterisks ($^{*}p < 0.05$, $^{***}p < 0.001$). **b** Same as **a**, but for normalized surface area (percent of cortical surface area; percentage values are out of 100). Note that there was a main effect of species ($p < 0.0001$), such that mcgs components were relatively larger in chimpanzees than in humans. There were no interactions with component. db dorsal branch, mb marginal branch; sb side branch.

$\eta2 = 0.16$), such that components of the mcgs are generally deeper in the left than right hemisphere (Fig. 9a). Third, there was a main effect of species ($F(1, 130) = 17.29$, $p < 0.0001$, $\eta2 = 0.12$) in which chimpanzee mcgs components were relatively deeper than humans (Fig. 9a). Fourth, there was an interaction between species and component ($F(2, 341) = 12.76$, $p < 0.0001$, $\eta2 = 0.07$). Post hoc analyses revealed that the db ($p < 0.0001$) and mb ($p = .03$) of the mcgs were relatively deeper in chimpanzees, whereas the sb was comparably deep between species ($p = 0.99$; Fig. 9a). Fifth, there was a three-way interaction among species, component, and hemisphere ($F(2, 341) = 5.58$, $p = 0.004$, $\eta2 = 0.03$). Post hoc analyses revealed that it was driven by i) the mb of the mcgs being relatively deeper in chimpanzees in the left hemisphere ($p = 0.03$), but not the right ($p = 0.32$; Fig. 9a) and ii) the species difference (i.e., chimpanzee>human) for the db being larger in the right hemisphere (estimate = 0.11, $p < 0.0001$) than the left (estimate = 0.04, $p = 0.02$).

In terms of surface area, an LME with component, hemisphere, and species on mcgs component as predictors revealed two findings. First, there was a main effect of component ($F(2, 341) = 971.27$, $p < 0.0001$, $\eta2 = 0.85$), such that the mb was larger than the db and sb ($ps < 0.0001$) and the db was larger than the sb ($p < 0.0001$; Fig. 9b). Second, there was a main effect of species ($F(1, 130) = 39.67$, $p < 0.0001$, $\eta2 = 0.23$) in which the mcgs components were all relatively larger in chimpanzees compared to humans (Fig. 9b). There were no species-related interactions ($ps > 0.16$).

## Discussion

By manually defining PMC sulci in 144 human and 120 chimpanzee (*Pan Troglodytes*) hemispheres, we show that the surface anatomy of PMC substantially differs between these two hominoid species along four sulcal metrics: i) the amount of PMC buried in sulci, ii) incidence/patterning, iii) depth, and iv) surface area (Fig. 10 summarizes the major differences in PMC sulcal morphology between chimpanzees and humans). For the amount of PMC buried in sulci, the amount in humans is larger than in chimpanzees. For sulcal incidence rates, half of PMC sulci are less present in chimpanzees than humans, whereas the other half are either more present in chimpanzees or equally present between

species (Fig. 10). Further, the prominent mcgs differs significantly between species (Fig. 10). For sulcal depth, the majority of PMC sulci are relatively shallower in chimpanzees compared to humans; however, a minority are relatively deeper in chimpanzees or equally deep in both species (Fig. 10). For sulcal surface area, the majority of PMC sulci are relatively smaller in chimpanzees compared to humans; however, a minority are relatively larger in chimpanzees or equally sized across species (Fig. 10). This variability is in stark contrast to previous work claiming similarities in PMC sulcal patterning between species:

"Overall, the medial aspect of the parietal lobe of the chimpanzee and other apes closely resembles the general appearance of the same structures in the human brain (Bailey et al., 1950)" [Cavanna and Trimble[76], pg. 565]

In the following sections, we discuss these findings in the context of the evolution of the cerebral cortex and the evolution of complex brain functions and behaviors, as well as discuss limitations and implications for future studies.

The present work adds to the growing literature in comparative neurobiology and paleoneurobiology classifying the presence/absence of sulci across species as a qualitative and quantitative metric to assess the evolution of the cerebral cortex. Such studies have revealed that although the sulcal patterning of primary sensory cortices more or less resembles one another across species[26], this relationship is far less consistent in association cortices. For example, while the sulcal organization of visual association cortex was comparable between every human and non-human hominoid hemisphere examined in previous work[19], the incidence of sulci in medial[18,20,21] and lateral[2,23,43] prefrontal cortex, as well as orbitofrontal cortex[73] was substantially different across species. Adding to the complexity, within each of these regions, differences in sulcal incidence rates were greater for some sulci compared to others—elucidating specific areas of cortex that are particularly expanded/more complex in humans. For example, sulcal incidence between humans and chimpanzees in the lateral prefrontal cortex is more consistent across species in the posterior middle frontal gyrus than anterior middle frontal gyrus[43]. Further, some sulci in the human lateral prefrontal cortex are not present in non-human hominoids[2,23]. As shown in the present study, although the PMC is generally more evolutionarily

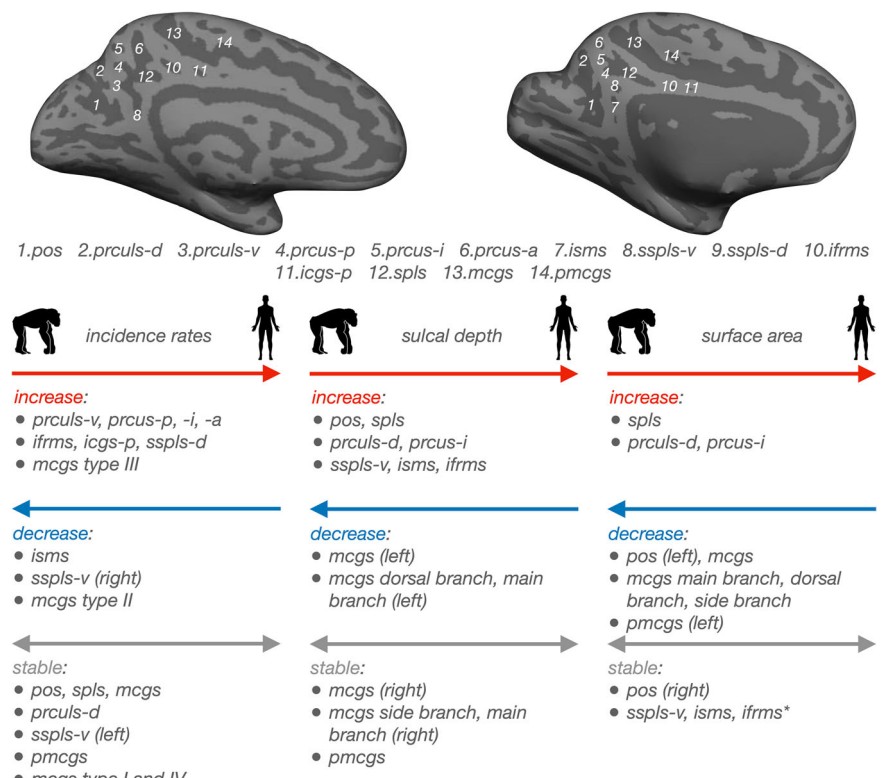

**Fig. 10 Summary of differences in PMC sulcal morphology between humans and chimpanzees.** Top: Inflated cortical surface reconstructions of the individual human (Left) and chimpanzee (Right) hemispheres shown in Fig. 2. Sulci: dark gray; Gyri: light gray. Individual sulci are numbered according to the key below. Note that the human hemisphere does not contain the isms (sulcus 7) or sspls-d (sulcus 9) and the chimpanzee hemisphere does not contain the prculs-v (sulcus 3) or sspls-d (sulcus 9). Bottom: Overview of differences in sulcal morphology between species. Position (right, left, both) of arrowheads indicates whether sulci increased (right), decreased (left), or remained stable (right and left) in each morphological feature between species. *Left:* incidence rates; *Middle:* sulcal depth; *Right:* surface area. ifrms data from ref. [24]. For sulci with species-related hemispheric effects, the specific hemisphere is shown next to the sulcus in parentheses. Chimpanzee and Human stock images were adapted from the Adobe Stock image database under the standard license (https://stock.adobe.com/license-terms).

expanded in humans[13,14,44–46], the differences in PMC sulcal morphology between humans and chimpanzees were heterogeneous — that is, not all sulci were less present, relatively smaller, and relatively shallower in chimpanzees compared to humans (Fig. 10).

Here, we consider four different underlying features that could contribute to this observed heterogeneity. First, the main border sulci (pos and mcgs) were relatively smaller and shallower in humans compared to chimpanzees (Fig. 10). This finding could be a consequence of the large increase in size, depth, and number of PrC sulci observed in humans compared to chimpanzees (Fig. 10). This observation is consistent with the classic compensation theory of cortical folding by Connolly[77,78], which qualitatively states that the depth and size of sulci are seemingly counterbalanced by those of their neighbors. In terms of the compensation theory then, in chimpanzees, the shallow, small (or even absent) precuneal sulci neighbor large and deep pos and mcgs (and the reverse in humans), such that the former "compensate" for the latter and in turn, make the overall degree of cortical folding approximately equal[79]. Second, PrC sulci were relatively larger in humans compared to chimpanzees, whereas PCC sulci were not (Fig. 10). This could be a consequence of the PrC not being topographically constrained along the vertical axis, in contrast to the PCC which is constrained superiorly by the cingulate/splenial sulci and inferiorly by the callosal sulcus. Recent empirical evidence by Bruner and colleagues supports this notion, finding that the PrC is the only area of PMC that spatially expands (in the longitudinal direction) between chimpanzees and

humans[13,14]. The majority of sulci in PrC and PCC were also relatively deeper in humans than chimpanzees, which could be due to the fact that both areas are not topographically constrained along this axis. Third, the decrease in isms presence in humans (Fig. 10) may be a consequence of changes in pos morphology in humans. For example, the pos intersects with the calcarine sulcus (e.g.,[70,80–83]) much more frequently in humans compared to chimpanzees[63–65,80,84,85]. The intersection of these two sulci, which is in the proximity of the isms, may have led to its decreased incidence rates in humans compared to chimpanzees. Fourth, the finding that ventral PCC sulci (sspls-v, isms) were more present or comparably present in chimpanzees and humans (Figs. 5, 10), is consistent with prior work showing that this area is one of the least evolutionarily expanded subregions of PMC[13,14,44–46].

In sum, these aforementioned results in PMC are consistent with complementary lines of work on cortical expansion in both paleoneurobiological[13,14] and in-vivo neuroimaging[44–46]. Nevertheless, it is an open question as to how consistent the relationship between species-related differences in expansion and sulcal morphology is across association cortices. There exists (at least) four options which can continue to be explored with further research: 1) increased cortical expanse with more sulci in humans compared to other species, 2) increased cortical expanse in humans with a comparable sulcal organization across species, 3) relatively similar sized cortical expanse with more sulci in humans compared to other species, and 4) relatively similar sized cortical expanse with a comparable sulcal organization across

species. For example, medial prefrontal cortex (BA 9) is a highly expanded region[44–46] with the appearance of new sulci in humans[18], which is consistent with Option 1. Nevertheless, the temporal lobe is more comparable between species than previously thought — both on the basis of paleoneurobiological expansion[86] and sulcal presence[19,87]), which is consistent with Option 4. As discussed in the previous paragraph, different subregions of PMC align differently with the four proposed options. Finally, considering that the present work only examined the PMC in chimpanzees, future work should also examine PMC sulcal morphology in additional species such as macaques, baboons, bonobos, gorillas, orangutans, and gibbons in order to build a larger picture regarding if/how the PMC changes along the primate phylogeny, with a particular emphasis on the presence/absence of the smaller, shallower, and more variable sulci[25].

The present findings also lay a foundation to examine the cognitive and functional role of PMC sulci in species beyond humans. Recent work shows that sulcal morphology relates to the appearance of complex behaviors in non-human hominoids[23,40,88,89]. For example, asymmetries in the depths of multiple sulci[23,40,88], as well as the presence of the paracingulate sulcus[40] and dorsal fronto-orbital sulcus pattern[23], relate to the production and use of attention-getting sounds by chimpanzees. Further, asymmetries in the depth of the inferior arcuate sulcus was related to gestural communication in baboons[89], as was the presence of the intralimbic sulcus in chimpanzees[40]. Thus, a goal for future work would be to relate the incidence rates and morphological features of PMC sulci to behavioral performance in non-human hominoids.

In conclusion, our findings provide insight regarding how PMC sulcal patterning and morphology differ between humans and our close relative: the chimpanzee. We not only uncover the presence of previously overlooked structures in human and chimpanzee PMC, but also show that the sulcal organization of PMC differs substantially between chimpanzees and humans along multiple metrics: percentage of sulci, sulcal presence, surface area, and depth. Future research can seek to further explore how the PMC sulcal patterning differs in humans relative to other non-human hominoids and non-human primates, as well as link the morphology of these structures to the emergence of complex behaviors, functional areas, and cortical networks.

## Materials and methods

### Participants

*Humans.* Data for the young adult human cohort analyzed in the present study were taken from the Human Connectome Project (HCP) database (https://www.humanconnectome.org/study/hcp-young-adult/overview). Here we used data from 72 randomly selected participants (36 females, 36 males, aged between 22 and 36). HCP consortium data were previously acquired using protocols approved by the Washington University Institutional Review Board. Informed consent was obtained from all participants. Here, we used the same participants used in our previous work in PMC identifying the ifrms for the first time[24].

*Chimpanzees.* 60 (37 female, 23 male, aged between 9 and 51) chimpanzee (*Pan Troglodytes*) anatomical T1 scans were chosen from the National Chimpanzee Brain Resource (www.chimpanzee.brain.org; supported by NIH grant NS092988). The chimpanzees were members of the colony housed at the Yerkes National Primate Research Center (YNPRC) of Emory University. All methods were carried out in accordance with YNPRC and Emory University's Institutional Animal Care and Use Committee (IACUC) guidelines. Institutional approval was obtained prior to the onset of data collection. Further data collection details are described in Keller et al.[6]. Here, we examined the same chimpanzees used in our prior work in PMC and other cortical expanses[19,24,43].

### Data acquisition

*Humans.* Anatomical T1-weighted (T1-w) MRI scans (0.8 mm voxel resolution) were obtained in native space from the HCP database. First, the images obtained from the scans were averaged. Then, reconstructions of the cortical surfaces of each participant were generated using FreeSurfer, a software used for processing and analyzing human brain MRI images (v6.0.0, surfer.nmr.mgh.harvard.edu). All

subsequent sulcal labeling and extraction of anatomical metrics were calculated from the cortical surface reconstructions of individual participants generated through the HCP's custom-modified version of the FreeSurfer pipeline[90].

*Chimpanzees.* Detailed descriptions of the scanning parameters have been described in Keller et al.[6], but we also describe the methods briefly here. Specifically, T1-weighted magnetization prepared rapid-acquisition gradient echo (MPRAGE) MR images were obtained using a Siemens 3T Trio MR system (TR = 2300 ms, TE = 4.4 ms, TI = 1100 ms, flip angle = 8, FOV = 200 mm) at YNPRC in Atlanta, Georgia. Before reconstructing the cortical surface, the T1 of each chimpanzee was scaled to the size of the human brain. As described in Hopkins et al.[91], within the FMRIB Software Library (FSL), the BET function was used to automatically strip away the skull, (2) the FAST function was used to correct for intensity variations due to magnetic susceptibility artifacts and radio frequency field inhomogeneities (i.e., bias field correction), and (3) the FLIRT function was used to normalize the isolated brain to the MNI152 template brain using a 7 degree of freedom transformation (i.e., three translations, three rotations, and one uniform scaling), preserved the shape of individual brains. Next, each T1 was segmented using FreeSurfer. The fact that the brains are already isolated, both bias-field correction and size-normalization, greatly assisted in segmenting the chimpanzee brain in FreeSurfer. Furthermore, the initial use of FSL also has the specific benefit, as mentioned above, of enabling the individual brains to be spatially normalized with preserved brain shape, and the values of this transformation matrix and the scaling factor were saved for later use.

### A brief explanation of sulcal terminology.
"Evolutionarily new" is used to describe small, shallow, and variable sulci because they are located either in association cortices that have expanded throughout evolution (e.g., lateral prefrontal cortex, medial parietal cortex, etc.[3,10,12–14,16,44–46]) or are unique to the hominoid brain (e.g., the fusiform gyrus[19,47]) and thus, absent in other primates such as macaques[25,69]. In some cases, non-human primates may contain small, superficial indentations in association cortices in a similar location as in humans that are considered spurs or dimples serving as "precursors of sulci that find full expression in the chimpanzee and human brains" [Amiez et al.[18], pg. 3].

These "evolutionarily new" sulci often correspond to tertiary sulci, which are located in association cortices and morphologically, are the smallest, shallowest and most variable cortical indentations (e.g.,[18,19,24,29,31,32,34,41,43,74,79,92–96]). In the present study, we use this latter definition for tertiary sulci. Nevertheless, classification of sulci as "tertiary" is classically based on their emergence in gestation[79,92–94,97] and these studies did not consider the timepoints of gestational emergence for newly discovered small, shallow, and variable sulci, which can be determined in future studies leveraging freely available fetal neuroimaging datasets.

### Manual sulcal labeling: all PMC sulci.
For this project, we manually defined 1619 sulci in 144 human hemispheres and 918 sulci in 120 chimpanzee (*Pan Troglodytes*) hemispheres.

*Humans.* For the present study, we re-assessed the 144 human hemispheres analyzed in our prior work[24]. Manual lines were drawn on the FreeSurfer *inflated* cortical surface to define sulci with tools in *tksurfer* based on the most recent schematics of sulcal patterning in PMC by Petrides[69], as well as by the pial and smooth surfaces of each individual as in our prior work[24,32,33,41]. In some cases, the precise start or end point of a sulcus can be difficult to determine on a surface[98]. Thus, using the inflated, pial, and smooth surfaces of each individual to inform our labeling allowed us to form a consensus across surfaces and clearly determine each sulcal boundary. For each hemisphere, the location of PMC sulci was identified by trained raters (E.H.W., S.A.M., J.K., B.P., T.H., L.A.G.) and confirmed by a trained neuroanatomist (K.S.W.).

In this process, we started with the large and deep sulci that bound PMC. Specifically, PMC is bounded posteriorly and anteriorly by the parieto-occipital sulcus (pos) and marginal ramus of the cingulate sulcus (mcgs), respectively. The splenial sulcus (spls) serves as a boundary between two subregions of PMC, the (superior) precuneus (PrC) and (inferior) posterior cingulate cortex (PCC), from one another[24,68]. In the present study, we also identified a previously unidentified sulcal component of the cingulate sulcus residing between the mcgs and paracentral sulcus[18,69] and below the paracentral fossa[69] in the paracentral lobule, which we term the premarginal branch of the cingulate sulcus (pmcgs). Although the pmcgs resides outside PMC (Fig. 2a, b; Supplementary Fig. 5)[24,48–50], since this sulcus demarcates the point at which the mcgs extends from the main body of the cingulate sulcus and prior work did not identify this structure (Supplementary Figs. 3, 4), we include it in the present study. See Supplementary Fig. 5 for examples of the pmcgs relative to cytoarchitectural regions outside of PMC as defined by multiple groups[99–101].

As shown in previous work[24], there are four consistent sulci within PrC: the dorsal precuneal limiting sulcus (prculs-d) and three precuneal sulci (posterior: prcus-p, intermediate: prcus-i, anterior: prcus-a). Within PCC, our prior work identified three small and shallow sulci[24]. The inframarginal sulcus (ifrms) is present in every human hemisphere inferior to the mcgs. Anterior to the ifrms, there is a variably present indentation termed the posterior intracingulate sulcus (icgs-p) based on the intracingulate sulcus nomenclature by Borne and

colleagues[98]. Posterior to the ifrms is the dorsal subsplenial sulcus (sspls-d) which is directly inferior to the main body of the spls.

In the present study, we identified three additional sulci not previously considered. The first sulcus is directly inferior to the posterior portion of the spls and more ventral along PCC—the ventral subsplenial sulcus (sspls-v) that is positioned underneath the sspls-d (when present). The second sulcus is posterior to prcus-p and inferior to the prculs-d—the ventral precuneus limiting sulcus (prculs-v). The third is a previously uncharted and lone indentation appearing within the isthmus of the cingulate gyrus, which we accordingly term the isthmus sulcus (isms). See Fig. 2a for 7 example human hemispheres with PMC sulci defined, and Supplementary Figure 1 for every hemisphere with sulcal labels.

*Chimpanzees.* Guided by recent in vivo criteria for defining PMC sulci in humans[24], we defined PMC sulci in chimpanzees. Prior work leveraging this same chimpanzee sample determined that chimpanzees variably possess an ifrms[24] and it is known that chimpanzees possess an mcgs, pos, and spls[63–65]. Therefore, in the present study, we determined whether or not chimpanzees possessed the pmcgs, as well as the five PrC sulci (prculs-d, prculs-v, prcus-p, prcus-i, prcus-a) and the four other PCC sulci (isms, sspls-v, sspls-d, icgs-p) residing within the bounds of the mcgs, pos, and spls in humans. As with humans, PMC sulci were defined in FreeSurfer using *tksurfer* tools, and for each hemisphere, the location of PMC sulci was confirmed by the same two-tiered process. See Fig. 2b for 7 example chimpanzee hemispheres with PMC sulci defined, and Supplementary Figure 2 for every hemisphere with sulcal labels.

**Manual sulcal labeling: mcgs patterns.** Linking to prior work by Bailey and colleagues[64] and Ono and colleagues[70], all 144 human and 120 chimpanzee inflated hemispheres were inspected by authors E.H.W., S.A.M., and K.S.W. to determine which of the four mcgs patterns was present in humans and chimpanzees: I) a main branch (mb) with no dorsal branch (db) or side branch (sb), II) mb with a db, III) mb with a sb, and IV) mb with both a db and sb.

**Calculating the amount of cortex buried in PMC across species.** To quantify the amount of cortex buried in PMC across individuals and species, we combined six regions in the Destrieux parcellation[66] corresponding to PMC: *G_cingul-Post-dorsal, G_cingul-Post-ventral, G_precuneus, S_cingul-Marginalis, S_parieto_occipital,* and *S_subparietal* (https://surfer.nmr.mgh.harvard.edu/fswiki/CorticalParcellation). These labels were converted from the Destrieux annotation into individual labels and combined into one "PMC ROI" FreeSurfer label with the *mri_annot2label* and *mri_mergelabels* functions in FreeSurfer (Fig. 1a). To quantify the areas of the cortex defined as sulci, we used the.sulc file (Fig. 1a)[67]. Depth values in the.sulc file are calculated based on how far removed a vertex is from what is referred to as a "mid-surface," which is determined computationally so that the mean of the displacements around this "mid-surface" is zero. Thus, generally, gyri have negative values, while sulci have positive values. To create a "sulci ROI" FreeSurfer label, we thresholded the.sulc file for all vertices with values > 0 with the *mri_binarize* function in Free-Surfer. To determine the percent of PMC composed of sulci, we calculated the overlap between the PMC ROI and sulci ROI with the Dice coefficient (Eq. 1)[24,41]:

$$DICE(X, Y) = \frac{2|X \cap Y|}{|X| + |Y|} \qquad (1)$$

where $X$ and $Y$ are the PMC ROI and sulci ROI, | | represents the number of elements in a set, and $\cap$ represents the intersection of two sets.

We then ran a linear mixed effects model (LME) with predictors of hemisphere and species, as well as their interaction terms, for percent overlap. Hemisphere and species were considered fixed effects. Hemisphere was nested within subjects. We also controlled for differences in brain size in the model (quantified as the total cortical surface area of the given hemisphere).

**Analyzing differences in sulcal incidence**
*PMC sulci.* We characterized the frequency of occurrence of each sulcus separately for left and right hemispheres. We first tested for broad differences in sulcal incidence in PrC and PCC separately using LMEs with predictors of hemisphere and species, as well as their interaction terms, for sulcal count. Hemisphere and species were considered fixed effects. Hemisphere was nested within subjects. Next, in line with prior work[18], for any sulcus that was not present in all hemispheres for either species, we tested the influence of species and hemisphere on the probability of a sulcus to be present with binomial logistic regression GLMs. For each statistical model, species (human, chimpanzee) and hemisphere (left, right), as well as their interaction, were included as factors for presence [0 (absent), 1 (present)] of a sulcus. To further probe differences in the subregions of PCC[49,102], we also conducted a follow-up GLM with PCC region [dorsal PCC (sspls-d, ifrms, icgs-p), ventral PCC (isms, sspls-v)], species (human, chimpanzee) and hemisphere (left, right), as well as their interaction, as factors for the presence of a sulcus.

Finally, to compare whether the incidence of the variable PMC sulci in chimpanzees related to one another, we ran binomial logistic regression GLMs for each variable PMC sulcus [0 (absent), 1 (present)] with the other sulci as factors, while also including an interaction with hemisphere for each sulcus. We iteratively dropped the sulcus that was the dependent variable as a factor from the next model to account for relationships already analyzed. Note that we excluded sulci with an

incidence rate of over 90% (prculs-d) and less than 15% (prculs-v, sspls-d, prcus-p, prcus-a, icgs-p) due to the very small sample size.

*Marginal ramus of the cingulate sulcus types.* We quantitatively determined whether the incidence rates of the four mcgs types differed by species, as well as between hemispheres for each species, with chi-squared ($\chi2$) tests.

**Quantification of sulcal morphology.** In the present study, we considered depth and surface area as these are two of the most defining morphological features of cortical sulci—especially in PMC (e.g.,[19,24,29–32,34,41,43,74,79,92–95,103–105]).

*Depth.* The depth of each sulcus was calculated in millimeters from each native cortical surface reconstruction. Raw values for sulcal depth were calculated from the sulcal fundus to the smoothed outer pial surface using a modified version of a recent algorithm for robust morphological statistics which builds on the FreeSurfer pipeline (Madan, 2019). As the chimpanzee surfaces were scaled prior to reconstruction, we report relative (normalized) depth values for the sulci of interest. For these metrics, within each species, depth was calculated relative to the deepest point in the cortex (i.e., the insula as in previous work[19,24,43]).

*Surface area.* Surface area (in square millimeters) was generated for each sulcus from the *mris_anatomical_stats* function in FreeSurfer[67,106]. Again, as in prior work[43], to address scaling concerns between species, we report surface area relative to the total cortical surface area of the given hemisphere.

**Morphological comparisons.** To assess whether the depth and surface area of PMC sulci differed between chimpanzees and humans, for both morphological features, we ran an LME with predictors of sulcus, hemisphere, and species, as well as their interaction terms. Species, hemisphere, and sulcus were considered fixed effects. Sulcus was nested within the hemisphere which was nested within subjects. For brevity, and considering that human PMC sulcal morphology has already been examined in prior work[24], we only report species-related effects in the main text for this set of analyses (i.e., not discussing main effects of sulcus). For these analyses we did not include the *ifrms* as our prior work[24] already conducted comparative morphological analyses on this sulcus in these two samples. Again, we excluded the sulci whose incidence rates were less than 15% in chimpanzees (prculs-v, sspls-d, prcus-p, prcus-a, icgs-p) from these analyses.

Finally, we repeated the prior analysis, exchanging the factor of PMC sulci for the mcgs branch (main branch, dorsal branch, side branch). As this is the first time these pieces have been quantitatively described, we report all effects in the main text.

**Statistics and reproducibility.** All statistical tests were implemented in R (v4.0.1) on the full human ($N = 72$, 144 hemispheres, 1619 sulci) and chimpanzee ($N = 60$, 120 hemispheres, 918 sulci) samples. LMEs were implemented with the *lme* function from *nlme* R package. ANOVA F-tests tests were applied to each GLM with the *anova* function from the built-in *stats* R package, from which results were reported. Effect sizes for the ANOVA effects are reported with the partial eta-squared (η2) metric and computed with the *eta_squared* function from the *effectsize* R package. GLMs were carried out with the *glm* function from the built-in *stats* R package. ANOVA $\chi2$ tests were applied to each GLM with the *Anova* function from the *car* R package, from which results were reported. Relevant post hoc analyses on ANOVA effects were computed with the *emmeans* and *contrast* functions from the *emmeans* R package (*p*-values adjusted with Tukey's method). Non-ANOVA $\chi2$ tests (for the mcgs type analysis) were carried out with the *chisq.test* function from the built-in *stats* R package. Follow-up post hoc pairwise comparisons on these $\chi2$ tests were implemented with the *chisq.multcomp* function from the *RVAideMemoire* R package.

**Reporting summary.** Further information on research design is available in the Nature Portfolio Reporting Summary linked to this article.

## Data availability
Processed sulcal data used for the present project are publicly available at GitHub (https://github.com/cnl-berkeley/stable_projects/tree/main/PosteromedialSulci_Chimpanzees) and Zenodo (https://doi.org/10.5281/zenodo.7938953)[107] repositories. The colorblind-friendly color schemes used in our figures were created using the toolbox available at https://davidmathlogic.com/colorblind/. Anonymized HCP neuroimaging data are available on ConnectomeDB (db.humanconnectome.org). Requests for further information should be directed to the Corresponding Author, Kevin Weiner (kweiner@berkeley.edu).

## Code availability
All original code used for the present project are publicly available at GitHub (https://github.com/cnl-berkeley/stable_projects/tree/main/PosteromedialSulci_Chimpanzees) and Zenodo (https://doi.org/10.5281/zenodo.7938953)[107] repositories.

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

## Acknowledgements

This research was supported by NSF CAREER Award 2042251 (Weiner) and NIH Grant R01MH129439 (Foster). Young adult neuroimaging and behavioral data were provided by the HCP, WU-Minn Consortium (Principal Investigators: David Van Essen and Kamil Ugurbil; NIH Grant 1U54-MH-091657) funded by the 16 NIH Institutes and Centers that support the NIH Blueprint for Neuroscience Research, and the McDonnell Center for Systems Neuroscience at Washington University. Chimpanzee data were provided by the National Chimpanzee Brain Resource (NIH Grant NS092988). We thank Willa Voorhies and Jacob Miller for their assistance developing the data collection pipeline used for this study, as well as Tyler Hallock and Lyndsey Aponik Gremillion for their previous assistance defining posteromedial sulci in humans.

## Author contributions

E.H.W. and K.S.W. designed research. E.H.W., S.A.M., J.P.K., B.J.P., B.L.F., and K.S.W. performed research. E.H.W., S.A.M., and K.S.W. analyzed data. E.H.W., S.A.M., and K.S.W. wrote the paper. All authors edited the paper and gave final approval before submission.

## Competing interests

The authors declare no competing interests.
