## [Peer Review File · Communications Biology]

Reviewers' comments:

Reviewer #1 (Remarks to the Author):

The present study is a qualitative and quantitative morphological sulcal study assessing whether and how the sulcal organization of the posteromedial cortex (PMC), including the posterior cingulate, retrosplenial and precuneal cortex, differs between humans (n=72) and chimpanzees (n=60). This is a significant paper showing that the PMC sulci present in humans are also present in chimpanzees and that differences between the 2 species lie mostly in the sulcal frequency of occurrence, sulcal depth, and surface area. My comments are below:

These results appear to contradict to neuroarcheology suggesting that PMC is the region that appears to have undergone the largest expansion in evolution (see several papers regarding PMC in Neanderthal from Emiliano Bruner's group). Indeed, such expansion should be associated with appearance of new sulci between chimpanzees and humans (as it has been shown in the medial area 9 for instance, Amiez et al. 2019). How the authors reconcile their data with this literature?

I have trouble to understand the idea of tertiary sulci and their systematic associations with being evolutionarily new. From my knowledge, primary, secondary, and tertiary sulci refer to their time of appearance in utero, not to their time of appearance during evolution. Second, it has been shown, for instance in the middle frontal cortex, that almost all main sulci that are found in humans are also observed in old-world monkeys or in chimpanzees the latest (Amiez et al. Nat Com 2019). As such, all these sulci are not evolutionary new. So, can the authors clarify this point?

The splenial sulcus is presented as bordering PMC, as the marginal ramus of the cingulate sulcus, and the parieto-occipital sulcus border it dorsally and ventrally respectively. However, this sulcus does not border PMC, it is at the center of it with roughly the posterior cingulate below it and the retrosplenial cortex above it.

From literature, PMC is located below the marginal ramus of the cingulate sulcus and above the parieto-occipital sulcus but pmcgs is clearly located above the marginal ramus of the cingulate sulcus. In which cytoarchitectonic area pmcgs falls? Why the authors interpret this sulcus as being part of PMC?

I suggest to remove the number of sulci manually labelled, this can be largely inferred and does not bring any pertinent additional information.

The authors do not mention any lateralization effect of PMC sulcal organization in both species. Is it because there is no effect?

Reviewer #2 (Remarks to the Author):

The authors performed a comparative analysis of the sulcal pattern in the posteromedial cortex (PMC) of the human and chimpanzee brains and quantified species differences in sulcal incidence, depth, and surface area, with a particular emphasis on the smaller, shallower, and relatively overlooked "evolutionarily new" dimples of the posterior cingulate, retrosplenial, and precuneal regions. They found that whereas most of the human PMC is buried in sulci (confirming previous observations), this does not apply to the chimpanzee brain. They found no interspecies differences concerning the incidence rates of sulci delimiting the PMC region. Conversely, sulci of the precuneal and posterior

cingulate (PCC) regions did show significant species-specific differences in incidence rates. Interestingly, most sulci identified by the authors within the precuneus were rarely identifiable in the chimpanzee brain, and the opposite holds true for sulci in the PCC. The authors also identified species differences in the surface area, though not the depth of the identified sulci. Importantly, these findings are discussed in the context of evolutionary changes in the forces driving cortical folding, as well as in and the evolution of higher cognitive functions, particularly in a key component of the default mode network. This highly detailed characterization of the similarities and differences in the sulcal pattern of the PMC in humans and chimpanzees provides a cornerstone for studies aiming to understand the evolution of structural-functional relationships.

Figure 10 is extremely helpful to keep track of the significant interspecies differences. The authors might consider introducing it at the beginning of the results section, instead of at the end. Also, given the number of sulci identified and of analyses performed, the authors may consider only describing statistically significant differences in the text, and providing the corresponding metrics in a table instead of in brackets in the main text. The reporting of χ^2 , df , η^2 , and p values after every corresponding significant difference (as well as the frequently repeated information that a Tukey's adjustment was performed) interrupts the flow of the sentence and makes the text currently very difficult to read.

Response to Reviewers

We thank the reviewers for their thoughtful and detailed comments on our paper. Below, we respond to comments raised by the reviewers and demonstrate how we have incorporated these suggestions into the manuscript. We believe that the changes implemented in the revision have improved the presentation of the results and the clarity of the manuscript.

Reviewer comments are in blue, and our responses are in black, with quotations from the text in gray.

Reviewer #1 (Remarks to the Author):

The present study is a qualitative and quantitative morphological sulcal study assessing whether and how the sulcal organization of the posteromedial cortex (PMC), including the posterior cingulate, retrosplenial and precuneal cortex, differs between humans (n=72) and chimpanzees (n=60). This is a significant paper showing that the PMC sulci present in humans are also present in chimpanzees and that differences between the 2 species lie mostly in the sulcal frequency of occurrence, sulcal depth, and surface area.

We thank the Reviewer for their positive feedback and for highlighting the significance of our paper.

My comments are below:

(1) These results appear to contradict to neuroarcheology suggesting that PMC is the region that appears to have undergone the largest expansion in evolution (see several papers regarding PMC in Neanderthal from Emiliano Bruner's group). Indeed, such expansion should be associated with appearance of new sulci between chimpanzees and humans (as it has been shown in the medial area 9 for instance, Amiez et al. 2019). How the authors reconcile their data with this literature?

We thank the Reviewer for the opportunity to clarify this concern. We cite the important work by Bruner and colleagues in the Discussion (Bruner et al., 2017 *Brain Struct Funct*; Bruner, 2018 *Brain Behav*). We further highlight that these previous paleoneurobiological findings align with the present findings in that the large evolutionary expansion of PMC as identified by Bruner and colleagues also co-occurs with multiple substantial differences in the sulcal organization of PMC in humans compared to chimpanzees as identified in the present study. For example, compared to the chimpanzee, human PMC sulci take up a greater extent of the region, are deeper and larger in

extent and frequency a majority of the time, and include sulci (prculs-v, sspls-d, mcgs type III pattern) unique to the human. Nevertheless, a surprising finding is that chimpanzees do have an increased incidence of two smaller and variable sulci compared to humans (isms and sspls-v). This finding aligns with prior work as these two sulci reside within a subregion of PMC that is not highly evolutionarily expanded (Hill et al., 2010 *Proc. Natl. Acad. Sci. U.S.A.*; Bruner et al., 2017 *Brain Struct Funct*; Bruner, 2018 *Brain Behav*).

In the **Discussion**, we now further clarify these points (Pgs. 22-23, Lines 451-481):

Recent empirical evidence by Bruner and colleagues supports this notion, finding that the PrC is the only area of PMC that spatially expands (in the longitudinal direction) between chimpanzees and humans^{13,14}. The majority of sulci in PrC and PCC were also relatively deeper in humans than chimpanzees, which could be due to the fact that both areas are not topographically constrained along this axis. Third, the decrease in isms presence in humans (**Fig. 10**) may be a consequence of changes in pos morphology in humans. For example, the pos intersects with the calcarine sulcus (e.g.,^{70,80-83}) much more frequently in humans compared to chimpanzees^{63-65,80,84,85}. The intersection of these two sulci, which is in the proximity of the isms, may have led to its decreased incidence rates in humans compared to chimpanzees. Fourth, the finding that ventral PCC sulci (sspls-v, isms) were more present or comparably present in chimpanzees and humans (**Figs. 5, 10**), is consistent with prior work showing that this area is one of the least evolutionarily expanded subregions of PMC^{13,14,44-46}.

In sum, these aforementioned results in PMC are consistent with complementary lines of work on cortical expansion in both paleoneurobiological^{13,14} and in-vivo neuroimaging⁴⁴⁻⁴⁶. Nevertheless, it is an open question as to how consistent the relationship between species-related differences in expansion and sulcal morphology are across association cortices. There exists (at least) four options which can continue to be explored with further research: 1) increased cortical expanse with more sulci in humans compared to other species, 2) increased cortical expanse in humans with a comparable sulcal organization across species, 3) relatively similar sized cortical expanse with more sulci in humans compared to other species, and 4) relatively similar sized cortical expanse with a comparable sulcal organization across species. For example, medial prefrontal cortex (BA 9) is a highly expanded region⁴⁴⁻⁴⁶ with the appearance of new sulci in humans¹⁸, which is consistent with Option 1. Nevertheless, the temporal lobe is more comparable between species than previously thought — both on the basis of paleoneurobiological expansion⁸⁶ and sulcal presence^{19,87}), which is consistent with Option 4. As discussed in the previous paragraph, different subregions of PMC align differently with the four proposed options. Finally, considering that the present work only examined the PMC in chimpanzees, future work should also examine PMC sulcal morphology in additional species such as macaques, baboons, bonobos, gorillas, orangutans, and gibbons in order to build a larger picture regarding if/how the PMC changes along the primate phylogeny, with a particular emphasis on the presence/absence of the smaller, shallower, and more variable sulci²⁵.

(2) I have trouble to understand the idea of tertiary sulci and their systematic associations with being evolutionarily new. From my knowledge, primary, secondary, and tertiary sulci refer to their time of appearance in utero, not to their time of appearance during evolution. Second, it has been shown, for instance in the middle frontal cortex, that almost all main sulci that are found in humans are also observed in old-world monkeys or in chimpanzees the latest (Amiez et al. Nat Com 2019). As such, all these sulci are not evolutionarily new. So, can the authors clarify this point?

We have added this text for clarification of these terms in the **Introduction** (Pg. 3, Lines 66-73):

Intriguingly, recent studies have identified “evolutionarily new” shallow sulci that have been linked to functional organization across a broad array of cognitive domains (e.g.,^{18,23,28–43}), several of which reflect cognitive abilities that are arguably unique to humans. We refer to these small, shallow, and variable sulci as “evolutionarily new” because they are located either in association cortices that have expanded throughout evolution (e.g., lateral prefrontal cortex, medial parietal cortex, etc.^{3,10,12–14,16,44–46}) or are unique to the hominoid brain and are absent in other primates (e.g., the fusiform gyrus^{19,47}; **Materials and Methods** for additional details on sulcal classification).

And the **Materials and Methods** (Pg. 26, Lines 546-560):

A brief explanation of sulcal terminology

“Evolutionarily new” is used to describe small, shallow, and variable sulci because they are located either in association cortices that have expanded throughout evolution (e.g., lateral prefrontal cortex, medial parietal cortex, etc.^{3,10,12–14,16,44–46}) or are unique to the hominoid brain (e.g., the fusiform gyrus^{19,47}) and thus, absent in other primates such as macaques^{25,69}. In some cases, non-human primates may contain small, superficial indentations in association cortices in a similar location as in humans that are considered spurs or dimples serving as “precursors of sulci that find full expression in the chimpanzee and human brains” [Amiez *et al.*¹⁸, pg. 3]. These “evolutionarily new” sulci often correspond to tertiary sulci, which are located in association cortices and morphologically, are the smallest, shallowest and most variable cortical indentations (e.g.,^{18,19,24,29,31,32,34,41,43,74,79,92–96}). In the present study, we use this latter definition for tertiary sulci. Nevertheless, classification of sulci as “tertiary” is classically based on their emergence in gestation^{79,92–94,97} and these studies did not consider the timepoints of gestational emergence for newly discovered small, shallow, and variable sulci, which can be determined in future studies leveraging freely available fetal neuroimaging datasets.

(3) The splenial sulcus is presented as bordering PMC, as the marginal ramus of the cingulate sulcus, and the parieto-occipital sulcus border it dorsally and ventrally respectively. However, this sulcus does not border PMC, it is at the center of it with roughly the posterior cingulate below it and the retrosplenial cortex above it.

We thank the reviewer for this clarification. Indeed, the splenial sulcus (spl_s) serves as a boundary separating two subregions of PMC, precuneal (PrC) and posterior cingulate (PCC) cortices. We now clarify this point in three places in the **Results** in which we write:

Pgs. 7-8, Lines 147-149:

We began by quantifying the incidence rates of PMC-related sulci in three groups: i) sulci that serve as the bounding perimeter of PMC or delimit PMC subregions (PrC, PCC), ii) sulci within PCC, and iii) sulci within PrC.

Pgs. 8-9, Lines 168-173:

Incidence rates of large and deep sulci that delimit PMC and its subregions do not differ across species

We identified two large and deep sulci serving as anterior and posterior bounds of PMC respectively, the marginal ramus of the cingulate sulcus (mcgs) and parieto-occipital sulcus (pos), as well as the splenial sulcus (spl_s), which separates PrC from PCC (**Fig. 2a, b; Materials and Methods**).

Pg. 9, Lines 187-192:

Figure 3. Incidence rates of sulci that delimit PMC and its subregions are comparable between humans and chimpanzees. Left: An inflated cortical surface reconstruction of an individual human (top) and chimpanzee (bottom) hemisphere with sulci that bound PMC and its subregions (PrC, PCC) outlined according to the legend at the top of the figure. Right: Bar plots visualizing incidence rates (percent of hemispheres) as a function of sulcus (x-axis), species (darker colors: human; lighter colors: chimpanzee), and hemisphere (left: left hemisphere; right: right hemisphere). Sulci are generally ordered posterior to anterior.

(4) From literature, PMC is located below the marginal ramus of the cingulate sulcus and above the parieto-occipital sulcus but pmcgs is clearly located above the marginal ramus of the cingulate sulcus. In which cytoarchitectonic area pmcgs falls? Why the authors interpret this sulcus as being part of PMC?

We agree that the pmcgs is located outside of both modern and classic definitions of PMC in humans and chimpanzees (e.g., see Parvizi et al., 2006 *Proc. Natl. Acad. Sci. U.S.A.*; Vickery et al., 2020 *eLife*; Willbrand et al., 2022 *Sci Adv*; Foster et al., 2023 *Nat Rev Neurosci*). Further, as suggested by Reviewer 1, it indeed resides outside the cytoarchitectural definition of PMC (e.g., Brodmann areas 7m, 23, 29, 30, 31) in modern and classic parcellations of the cortex. We have added a new Supplementary Figure 5 to show that the pmcgs does reside outside cytoarchitectural definitions of PMC: in areas 4a (Amunts et al., 2020 *Science*), 4 (Brodmann 1909), or FA (Von Economo and Koskinas, 1925).

Supplementary Figure 5. Premarginal branch of the cingulate sulcus (pmcgs) resides cytoarchitecturally outside of PMC. An example inflated human left hemisphere with the pmcgs (yellow) identified relative to different cytoarchitectural definitions within its proximity: Area 4 (left; blue outline) defined by Brodmann¹², Area 4a (PreCg) (middle; purple outline) as defined by observer-independent methods by Amunts *et al.*¹³, and Area FA (right; green outline) as defined by Von Economo and Koskinas¹⁴. In each case, the pmcgs resides outside of cytoarchitectonic PMC regions. This relationship also extends to chimpanzees¹⁰. Note that the areas around Area 4a have yet to be charted using modern, observer-independent techniques¹³.

Additionally, this relationship also holds in chimpanzee PMC parcellations [for reference see image from Bailey *et al.* (1950) below]. This point is touched on in the caption of Supplementary Figure 5 above.

Premarginal branch of the cingulate sulcus (pmcgs) resides cytoarchitecturally outside of PMC in chimpanzees. Schematic from Bailey *et al.* (1950) with the general PMC expanse outlined in red and the putative pmcgs location in black.

Nevertheless, in the present work, we included the pmcgs as it was a previously undocumented sulcal feature of the cgs that resides near to one of the larger and consistent PMC sulci (mcgs). Therefore, it is also appropriate to have considered this sulcus as part of assessing the morphology of the mcgs and what proximal features should be included/distinguished from it.

We have edited the **Results** to clarify this point (Pg. 9, Lines 174-182):

We also identified a consistent sulcus just anterior to the mcgs (**Fig. 2a, b**). As it is common practice to refer to consistent sulci as “pre” or “post” if they are located in front of or behind other prominent sulci (e.g., precentral, postcentral, and central sulci), we refer to this sulcus as the premarginal branch of the cingulate sulcus (pmcgs). When present, the pmcgs is located just under the paracentral fossa and serves as the point where the mcgs breaks from the cingulate sulcus (cgs) proper. Even though the pmcgs is technically located outside of what is considered PMC, we include it here as (to our knowledge), its consistency across hemispheres and species has not yet been documented until the present work (**Materials and Methods** for additional information regarding the pmcgs; Supplementary Fig. 5).

We have also edited the **Materials and Methods** to clarify this point (Pgs. 27-28, Lines 579-587):

In the present study, we also identified a previously unidentified sulcal component of the cingulate sulcus residing between the mcgs and paracentral sulcus^{18,69} and below the paracentral fossa⁶⁹ in the paracentral lobule, which we term the premarginal branch of the cingulate sulcus (pmcgs). Although the pmcgs resides outside PMC (**Fig. 2a, b**; Supplementary Fig. 5)^{24,48-50}, since this sulcus demarcates the point at which the mcgs extends from the main body of the cingulate sulcus and prior work did not identify this structure (Supplementary Figs. 3, 4), we include it in the present study. See Supplementary Figure 5 for examples of the pmcgs relative to cytoarchitectural regions outside of PMC as defined by multiple groups⁹⁹⁻¹⁰¹.

(5) I suggest to remove the number of sulci manually labeled, this can be largely inferred and does not bring any pertinent additional information.

We now only report the number of sulci manually labeled two times, once in the Abstract and once in the Materials and Methods.

In the **Abstract** (Pg. 2, Lines 47-49), we write:

To fill this gap in knowledge, we first manually defined over 2500 PMC sulci in 120 chimpanzee (*Pan Troglodytes*) hemispheres and 144 human hemispheres.

In the **Materials and Methods** (Pg. 27, Lines 563-564), we write:

For this project, we manually defined 1619 sulci in 144 human hemispheres and 918 sulci in 120

chimpanzee (*Pan Troglodytes*) hemispheres.

(6) The authors do not mention any lateralization effect of PMC sulcal organization in both species. Is it because there is no effect?

There were a handful of lateralization effects, which are mentioned throughout the **Results** section, when significant. For clarity, these lateralization effects are also summarized in Figure 10 (Pg. 20, Lines 409-418). We now highlight the lateralization effects in the caption to emphasize the presence of hemispheric asymmetries:

Figure 10. Summary of differences in PMC sulcal morphology between humans and chimpanzees. Top: Inflated cortical surface reconstructions of the individual human (Left) and chimpanzee (Right) hemispheres shown in Figure 2. Sulci: dark gray; Gyri: light gray. Individual sulci are numbered according to the key below. Note that the human hemisphere does not contain the isms (sulcus 7) or sspls-d (sulcus 9) and the chimpanzee hemisphere does not contain the prculs-v (sulcus 3) or sspls-d (sulcus 9). Bottom: Overview of differences in sulcal morphology between species. Position (right, left, both) of arrowheads indicates whether sulci increased (right), decreased (left), or remained stable (right and left) in each morphological feature between species. *Left:* incidence rates; *Middle:* sulcal depth; *Right:* surface area. ifrms data from²⁴. For sulci with

species-related hemispheric effects, the specific hemisphere is shown next to the sulcus in parentheses.

For the sulci not included in Figure 10 or the caption of Figure 10, there were no significant hemispheric asymmetries. We have added text at the beginning of the **Results** to emphasize this. On Pg. 8, Lines 156-166, we write:

We described across-species comparisons for each group in turn below using logistic regression generalized linear models (GLMs) with species (human, chimpanzee) and hemisphere (left, right), as well as their interaction, as factors for sulcal presence. Afterwards, we compared the depth and surface area of PMC sulci between species using LMEs with species (human, chimpanzee), sulcus (PMC sulci), and hemisphere (left, right), as well as their interaction, as factors. Finally, we repeat these analyses on the incidence and morphology of the marginal ramus of the cingulate sulcus — a prominent sulcal landmark in PMC^{24,55,64,68–72} that, contrary to previous studies, differs substantially between species, which we show here. In all analyses, observed hemispheric asymmetries were not significant ($ps > .05$), unless explicitly stated. Further, for all post hoc comparisons conducted, p -values were corrected with Tukey's methods (**Materials and Methods** for additional analysis details).

Reviewer #2 (Remarks to the Author):

The authors performed a comparative analysis of the sulcal pattern in the posteromedial cortex (PMC) of the human and chimpanzee brains and quantified species differences in sulcal incidence, depth, and surface area, with a particular emphasis on the smaller, shallower, and relatively overlooked “evolutionarily new” dimples of the posterior cingulate, retrosplenial, and precuneal regions.

They found that whereas most of the human PMC is buried in sulci (confirming previous observations), this does not apply to the chimpanzee brain. They found no interspecies differences concerning the incidence rates of sulci delimiting the PMC region. Conversely, sulci of the precuneal and posterior cingulate (PCC) regions did show significant species-specific differences in incidence rates. Interestingly, most sulci identified by the authors within the precuneus were rarely identifiable in the chimpanzee brain, and the opposite holds true for sulci in the PCC. The authors also identified species differences in the surface area, though not the depth of the identified sulci.

Importantly, these findings are discussed in the context of evolutionary changes in the forces driving cortical folding, as well as in and the evolution of higher cognitive functions, particularly in a key component of the default mode network.

This highly detailed characterization of the similarities and differences in the sulcal pattern of the PMC in humans and chimpanzees provides a cornerstone for studies aiming to understand the evolution of structural-functional relationships.

We thank the Reviewer for highlighting the significance of our work and its potential to serve as a cornerstone for future studies.

(1) Also, given the number of sulci identified and of analyses performed, the authors may consider only describing statistically significant differences in the text, and providing the corresponding metrics in a table instead of in brackets in the main text. The reporting of χ^2 , df, η^2 , and p values after every corresponding significant difference (as well as the frequently repeated information that a Tukey's adjustment was performed) interrupts the flow of the sentence and makes the text currently very difficult to read.

We thank Reviewer 2 for bringing up this point. We have taken multiple steps to consolidate statistical effects, which we believe have improved the clarity of the **Results**.

First, we have removed the mention of Tukey corrections for each analysis. Instead, we have added text in the **Results** and **Materials and Methods** to highlight the corrections being made for each analysis:

Results (Pg. 8, Lines 164-166):

Further, for all post hoc comparisons conducted, *p*-values were corrected with Tukey's methods (**Materials and Methods** for additional analysis details).

Materials and Methods (Pg. 33, Lines 712-714):

Relevant post hoc analyses on ANOVA effects were computed with the *emmeans* and *contrast* functions from the *emmeans* R package (*p*-values adjusted with Tukey's method).

Second, we consolidated the lack of hemispheric effects to one sentence at the beginning of the **Results** (also based on a comment from Reviewer 1; Pg. 8, Lines 163-164):

In all analyses, observed hemispheric asymmetries were not significant ($ps > .05$), unless explicitly stated.

Third, we have generally consolidated the statistics reported in parentheses. For example, we have shortened the degrees of freedom text for χ^2 tests (df = #) and instead, include them after the test statistic, $\chi^2(df)$, as done for F-tests.

(2) Figure 10 is extremely helpful to keep track of the significant interspecies differences. The authors might consider introducing it at the beginning of the results section, instead of at the end.

After re-reviewing the manuscript and the revisions suggested by Reviewer 1, we believe that moving Figure 10 will disrupt the flow of the manuscript — both based on prior evolutionary sulcal work that also included a review of the results at the end of the Results/beginning of the Discussion (e.g., see Amiez et al., 2019 *Nat Commun*) and because it aligns with the first paragraph of the **Discussion** (Pgs. 19-20, Lines 387-408) that summarizes our main results reported in the paper. However, we see the utility of providing an overview of the primary findings early in the manuscript, and so we have added additional text to the last paragraph of the **Introduction** (Pg. 4, Lines 87-99):

While it is known that the larger (primary) sulci within PMC are present in chimpanzees^{63–65} and the inframarginal sulcus — a newly uncovered smaller PMC sulcus — is variably present in chimpanzees²⁴, the phylogenetic emergence of a majority of recently clarified PMC sulci²⁴ has yet to be compared between chimpanzees and humans. Therefore, in the present study, we comprehensively examined the PMC sulcal patterning between humans and chimpanzees using cortical surface reconstructions as in our prior work^{19,24,43}. Our analyses were guided by three main questions. First, does the amount of PMC buried in sulci differ between humans and chimpanzees? Second, do the incidence rates of PMC sulci differ between species? Third, do the primary morphological features of these structures (i.e., depth and surface area) differ between species? Here, we uncovered four new sulci, and quantitatively identified species differences in incidence rates, depth, and surface area. Interestingly, some PMC sulci are more common in humans and others, in chimpanzees. Further, we found that the prominent and consistent marginal ramus of the cingulate sulcus differs significantly between species.

REVIEWERS' COMMENTS:

Reviewer #1 (Remarks to the Author):

The authors responded to all my concerns. This paper is convincing and bring important information. As such, I recommend the paper for publication.

Reviewer #2 (Remarks to the Author):

All my comments have been satisfactorily addressed by the authors, and I recommend that the article be accepted for publication

Response to Reviewers

We thank the reviewers for their thoughtful and detailed comments on our paper. Below, we respond to comments raised by the reviewers and demonstrate how we have incorporated these suggestions into the manuscript. We believe that the changes implemented in the revision have improved the presentation of the results and the clarity of the manuscript.

Reviewer comments are in blue, and our responses are in black, with quotations from the text in gray.

REVIEWERS' COMMENTS:

Reviewer #1 (Remarks to the Author):

The authors responded to all my concerns. This paper is convincing and bring important information. As such, I recommend the paper for publication.

We thank the reviewer for taking the time, care, and effort to review our manuscript and are pleased to hear that they approve our manuscript for publication.

Reviewer #2 (Remarks to the Author):

All my comments have been satisfactorily addressed by the authors, and I recommend that the article be accepted for publication

We thank the reviewer for taking the time, care, and effort to review our manuscript and are pleased to hear that they approve our manuscript for publication.